# Assessing the Functional Redundancy between P-gp and BCRP in Controlling the Brain Distribution and Biliary Excretion of Dual Substrates with PET Imaging in Mice

**DOI:** 10.3390/pharmaceutics13081286

**Published:** 2021-08-18

**Authors:** Irene Hernández-Lozano, Severin Mairinger, Alexander Traxl, Michael Sauberer, Thomas Filip, Johann Stanek, Claudia Kuntner, Thomas Wanek, Oliver Langer

**Affiliations:** 1Department of Clinical Pharmacology, Medical University of Vienna, 1090 Vienna, Austria; irene.hernandezlozano@meduniwien.ac.at (I.H.-L.); severin.mairinger@meduniwien.ac.at (S.M.); 2Preclinical Molecular Imaging, AIT Austrian Institute of Technology GmbH, 2444 Seibersdorf, Austria; alexander.traxl@gmail.com (A.T.); michael.sauberer@meduniwien.ac.at (M.S.); thomas.filip@meduniwien.ac.at (T.F.); johann.stanek@meduniwien.ac.at (J.S.); claudia.kuntner-hannes@meduniwien.ac.at (C.K.); thomas.wanek@meduniwien.ac.at (T.W.); 3Department of Biomedical Imaging and Image-Guided Therapy, Medical University of Vienna, 1090 Vienna, Austria; 4Center of Biomedical Research, Medical University of Vienna, 1090 Vienna, Austria

**Keywords:** P-glycoprotein, breast cancer resistance protein, functional redundancy, liver, brain, PET imaging, pharmacokinetic modeling

## Abstract

P-glycoprotein (P-gp) and breast cancer resistance protein (BCRP) are co-localized at the blood–brain barrier, where they display functional redundancy to restrict the brain distribution of dual P-gp/BCRP substrate drugs. We used positron emission tomography (PET) with the metabolically stable P-gp/BCRP substrates [^11^C]tariquidar, [^11^C]erlotinib, and [^11^C]elacridar to assess whether a similar functional redundancy as at the BBB exists in the liver, where both transporters mediate the biliary excretion of drugs. Wild-type, *Abcb1a/b^(−/−)^*, *Abcg2^(−/−)^*, and *Abcb1a/b^(−/−)^Abcg2^(−/−)^* mice underwent dynamic whole-body PET scans after i.v. injection of either [^11^C]tariquidar, [^11^C]erlotinib, or [^11^C]elacridar. Brain uptake of all three radiotracers was markedly higher in *Abcb1a/b^(−/−)^Abcg2^(−/−)^* mice than in wild-type mice, while only moderately changed in *Abcb1a/b^(−/−)^* and *Abcg2^(−/−)^* mice. The transfer of radioactivity from liver to excreted bile was significantly lower in *Abcb1a/b^(−/−)^Abcg2^(−/−)^* mice and almost unchanged in *Abcb1a/b^(−/−)^* and *Abcg2^(−/−)^* mice (with the exception of [^11^C]erlotinib, for which biliary excretion was also significantly reduced in *Abcg2^(−/−)^* mice). Our data provide evidence for redundancy between P-gp and BCRP in controlling both the brain distribution and biliary excretion of dual P-gp/BCRP substrates and highlight the utility of PET as an upcoming tool to assess the effect of transporters on drug disposition at a whole-body level.

## 1. Introduction

The adenosine triphosphate-binding cassette (ABC) transporters P-glycoprotein (P-gp, encoded in humans by the *ABCB1* gene and in rodents by the *Abcb1a* and *Abcb1b* genes, and breast cancer resistance protein (BCRP, encoded in humans by the *ABCG2* gene and in rodents by the *Abcg2* gene) are two important efflux transporters that are widely expressed throughout the body [1]. These two membrane transporters have a broad and largely overlapping substrate spectrum, including a variety of clinically used drugs, such as most molecularly targeted anticancer drugs [2,3]. At the luminal membrane of brain capillary endothelial cells, which form the blood–brain barrier (BBB), P-gp and BCRP play a vital defense role by limiting the distribution of their substrates into the brain by efflux transport from the endothelial cells back into the blood [4]. Several studies have provided evidence for functional redundancy between P-gp and BCRP in limiting the brain distribution of dual P-gp/BCRP substrates [2,3,4,5,6]. In absence of either P-gp alone or BCRP alone, the remaining transport capacity of the other transporter usually suffices to restrict the brain access of dual substrates so that dual P-gp/BCRP substrates only gain unrestricted brain access when both transporters are absent or inhibited [4]. Using individual and combined *Abcb1a/b* and *Abcg2* knockout mice, this functional redundancy between P-gp and BCRP at the BBB has been demonstrated for several dual P-gp/BCRP substrates, such as lapatinib, erlotinib, flavopiridol, and mitoxantrone [5,6]. There is also evidence that a similar functional redundancy between P-gp and BCRP exists at the human BBB [7]. In addition, P-gp and BCRP may become overexpressed in certain tumors, in which they can contribute to multidrug resistance by limiting cellular entry of anticancer drugs [8].

Apart from the BBB, P-gp and BCRP are also expressed in excretory organs, such as the liver, the small intestine, and the kidneys [1]. At the canalicular (bile-facing) membrane of hepatocytes, P-gp and BCRP have been shown to mediate the excretion of drugs and their metabolites into bile [1]. However, it is currently incompletely understood whether the P-gp/BCRP functional redundancy observed at the BBB also extends to the biliary excretion of dual P-gp/BCRP substrates.

Positron emission tomography (PET) is a non-invasive nuclear imaging method that enables measurement of the pharmacokinetics (PK) of radiolabeled drug molecules in different tissues and organs of the body [9]. Due to the limited field of view of most currently available clinical PET scanners, dynamic PET imaging in humans has thus far been mainly limited to a short axial segment of the body (approximately 20 cm). On the other hand, small-animal PET imaging in mice offers the possibility to simultaneously and dynamically measure the whole-body disposition of radiolabeled molecules. In combination with suitable PK models, quantitative parameters can be obtained that can be related to membrane transporter activity in different organs [10]. The carbon-11 (^11^C)-labeled third-generation P-gp inhibitors [^11^C]tariquidar and [^11^C]elacridar and the tyrosine kinase inhibitor [^11^C]erlotinib are dual P-gp and BCRP substrates [6,7,11] and have been used to assess the activity of P-gp and BCRP at the human and mouse BBB [7,11,12,13,14,15,16,17]. All three radiotracers are predominantly excreted via the hepatobiliary route [18,19].

The aim of this study was to assess whether the functional redundancy between P-gp and BCRP observed in mice for restricting the brain distribution of [^11^C]tariquidar, [^11^C]erlotinib, and [^11^C]elacridar also occurs with respect to their biliary excretion. We used previously published data sets, in which wild-type, *Abcb1a/b^(−/−)^*, *Abcg2^(−/−)^*, and *Abcb1a/b^(−/−)^Abcg2^(−/−)^* mice had been scanned with all three radiotracers [11,16]. Here, we exploited the ability of small-animal PET to perform dynamic whole-body imaging in mice, which enabled us to obtain quantitative PK parameters describing both the brain distribution and the biliary excretion of the three radiotracers in the four investigated mouse strains.

## 2. Materials and Methods

The data sets used for this study were previously published by Bankstahl et al. and Traxl et al. [11,16]. All animal experiments were approved by the national authorities (Amt der Niederösterreichischen Landesregierung), and all study procedures were performed in accordance with the European Communities Council Directives of 24 November 1986 (86/609/EEC) and 22 September 2010 (2010/63/EU).

### 2.1. Radiotracer Synthesis and Formulation

[^11^C]Tariquidar, [^11^C]erlotinib, and [^11^C]elacridar were synthesized as previously reported [20,21,22]. For intravenous (i.v.) injection into animals, [^11^C]tariquidar and [^11^C]elacridar were formulated in a mixture of 0.9% aqueous saline/ethanol/polyethylene glycol 300 (50:15:35, *v*/*v*/*v*) [11] and [^11^C]erlotinib was formulated in 0.1 mM hydrochloric acid in physiologic saline [16].

### 2.2. PET Imaging

Female wild-type, *Abcb1a/b^(−/−)^*, *Abcg2^(−/−)^*, and *Abcb1a/b^(−/−)^Abcg2^(−/−)^* mice with a FVB genetic background underwent under isoflurane/oxygen anesthesia 60 min dynamic PET scans after i.v. injection of either [^11^C]tariquidar (32 ± 9 MBq, corresponding to 0.20 ± 0.05 µg of unlabeled tariquidar), [^11^C]erlotinib (27 ± 8 MBq, corresponding to 0.79 ± 0.39 µg of unlabeled erlotinib), or [^11^C]elacridar (36 ± 8 MBq, corresponding to 0.19 ± 0.05 µg of unlabeled elacridar) using a microPET Focus 220 scanner (Siemens Medical Solutions, Knoxville, TN, USA) as previously described [11,16]. At the time of the experiments, wild-type mice weighed 23.1 ± 1.6 g, *Abcb1a/b^(−/−)^* mice 21.9 ± 2.1 g, *Abcg2^(−/−)^* mice 21.2 ± 3.1 g, and *Abcb1a/b^(−/−)^Abcg2^(−/−)^* mice 23.5 ± 2.7 g.

### 2.3. Analysis of PET Data

Images were analyzed with the software AMIDE [23]. Volumes of interest for the left ventricle of the heart (image-derived arterial blood curve), brain, liver, and intestine (representing all visible intestinal radioactivity) were manually outlined on the reconstructed PET images. Time activity curves (TACs) obtained from the selected volumes of interest were expressed as percent of injected dose per milliliter (%ID/mL), except for the intestine, for which radioactivity was expressed as %ID by multiplying the image-derived radioactivity concentration by the selected volume of interest. From the TACs, the area under the curve (AUC) was calculated using Prism 8 (GraphPad Software, La Jolla, CA, USA). In order to assess the brain distribution of [^11^C]tariquidar, [^11^C]erlotinib, and [^11^C]elacridar, we applied graphical non-compartmental analysis approaches. The initial uptake clearance of radioactivity from blood into brain (CL_uptake,brain_, mL/min/g tissue) was estimated using integration plot analysis [24,25] and the following equation:(1)$$\frac{X_{t,\mathrm{brain}}}{C_{t,\mathrm{blood}}}={\mathrm{CL}}_{\mathrm{uptake},\mathrm{brain}}\times \frac{{\mathrm{AUC}}_{0–t,\mathrm{blood}}}{C_{t,\mathrm{blood}}}+V_{E,\mathrm{brain}}$$
where X_t,brain_ is the amount of radioactivity per gram brain tissue at time t, C_t,blood_ is the radioactivity concentration in the blood at time t (image-derived from the left ventricle of the heart), AUC_0-t,blood_ is the area under the blood curve from time 0 to time t, and V_E,brain_ represents the capillary space as well as the rapid adsorption/binding to the vascular surface in the brain [25]. CL_uptake,brain_ is the slope (from 0.5 to 4.5 min) of a plot of X_t,brain_/C_t,blood_ versus AUC_0–t,blood_/C_t,blood_ and is estimated by performing linear regression analysis. In addition, the total brain distribution volume (V_T,brain_) was calculated with Logan graphical analysis [26]. V_T,brain_ equals the brain-to-blood concentration ratio at steady state. While this methodology has already been used to analyze the [^11^C]erlotinib data set [16], the brain distribution of [^11^C]tariquidar and [^11^C]elacridar has previously only been described in terms of brain radioactivity concentrations and brain-to-blood concentration ratios at 25 min after radiotracer injection [11].

In addition, the intestinal clearance of radioactivity with respect to the blood concentration (CL_intestine,blood_, mL/min) was calculated by dividing the total amount of radioactivity in the intestine at the end of the PET scan by AUC_blood_.

### 2.4. Pharmacokinetic Modeling

The [^11^C]tariquidar data set has already been previously analyzed with a three-compartment model [27,28], which was slightly modified in this work (Figure 1) and applied to all three radiotracers in order to estimate the PK parameters defining the transfer of radioactivity between the compartments. CL_1_ (mL/min) represents the hepatic uptake clearance, k_2_ (min^−1^) is the rate constant describing the transfer of radioactivity from liver into the sink compartment (blood), and k_3_ (min^−1^) is the rate constant describing the transfer of radioactivity from liver into excreted bile. The model assumes that all radioactivity in the intestine represents excreted bile and that direct secretion from blood into the intestine was negligible over the short duration of the PET scan. For this assumption to be confirmed, it would be necessary to examine bile duct-cannulated mice, which is technically challenging and was not performed in the present study. The radioactivity amount in the liver was corrected for the fraction of blood in the liver (≈0.25). In addition, the model accounts for radiotracer delivery via the portal vein and the hepatic artery. The radiotracer concentration in the hepatic artery was assumed to correspond to the image-derived arterial blood curve (obtained from the volume of interest placed in the left ventricle of the heart), while the concentration in the portal vein was mathematically estimated from the arterial blood and the PET data during the modeling process as previously described [28]. For modeling purposes, data were expressed in megabecquerel per milliliter (MBq/mL) for blood and liver, and in MBq for the intestine.

The main difference between the already published liver model [28] and the one used in this study is related to the implementation of the input function. In the present study, the input function used was the mathematically estimated blood concentration-time curve, while the previous model used the amount of radioactivity in the hepatic sinusoids as an input function (by multiplying the mathematically estimated blood concentration curve by the volume occupied by the hepatic sinusoids—approximately 25% of the liver volume). This change leads to a different PK parameter defining the radioactivity uptake, CL_1_ (mL/min) instead of k_1_ (min^−1^) [28].

### 2.5. Statistical Analysis

Statistical analysis was performed using Prism 8. Differences in PK parameters between mouse groups were assessed by one-way ANOVA followed by a Dunnett’s multiple comparison test against the wild-type group. The level of statistical significance was set to a *p*-value of less than 0.05. All values are given as mean ± standard deviation (SD).

## 3. Results

### 3.1. Functional Redundancy between P-gp and BCRP in Controlling the Brain Distribution of [^11^C]tariquidar, [^11^C]erlotinib, and [^11^C]elacridar

Wild-type, *Abcb1a/b^(−/−)^*, *Abcg2^(−/−)^*, and *Abcb1a/b^(−/−)^Abcg2^(−/−)^* mice underwent dynamic PET scans with either [^11^C]tariquidar, [^11^C]erlotinib, or [^11^C]elacridar. Mean TACs in blood (image-derived blood curve), brain, liver, and intestine are shown in Appendix A for [^11^C]tariquidar, in Appendix A for [^11^C]erlotinib, and in Appendix A for [^11^C]elacridar. In Appendix A, AUC values of the TACs in the brain and other examined tissues are given for all three radiotracers for the four mouse strains. As previously reported [11,16], brain radioactivity uptake was very low in wild-type mice for each radiotracer (Figure 2, Appendix A). Brain radioactivity uptake in *Abcb1a/b^(−/−)^* mice and in *Abcg2^(−/−)^* mice was comparably low as in wild-type mice, while a considerably higher brain uptake was observed in *Abcb1a/b^(−/−)^Abcg2^(−/−)^* mice for [^11^C]tariquidar and [^11^C]elacridar and to a lesser extent also for [^11^C]erlotinib.

Integration plot analysis was used to estimate the initial uptake clearance of radioactivity from blood into brain (CL_uptake,brain_, Figure 3, Table 1). CL_uptake,brain_ was significantly increased in *Abcb1a/b^(−/−)^Abcg2^(−/−)^* mice for all three radiotracers (by 7.4-fold for [^11^C]tariquidar, by 4.6-fold for [^11^C]erlotinib, and by 6.2-fold for [^11^C]elacridar), while only moderate and not significant changes were observed in *Abcb1a/b^(−/−)^* mice and *Abcg2^(−/−)^* mice as compared to the wild-type group. Similarly, the total brain distribution volume (V_T,brain_) was significantly higher in *Abcb1a/b^(−/−)^Abcg2^(−/−)^* mice than in wild-type mice, and not significantly changed in *Abcb1a/b^(−/−)^* mice and *Abcg2^(−/−)^* mice for all three radiotracers (Table 1).

### 3.2. Functional Redundancy between P-gp and BCRP in Mediating the Biliary Excretion of [^11^C]tariquidar, [^11^C]erlotinib, and [^11^C]elacridar

Representative PET summation images of the abdominal region obtained after i.v. injection of either [^11^C]tariquidar, [^11^C]erlotinib, or [^11^C]elacridar in the different mouse strains are shown in Figure 4. For each radiotracer, appreciable excretion of radioactivity into the intestine occurred in wild-type mice over the duration of the PET scan (Figure 4, Appendix A). At the end of the PET scan, the total amount of radioactivity in the intestine of wild-type mice was 17.2 ± 3.1 %ID for [^11^C]tariquidar (Appendix A), 60.8 ± 6.6 %ID for [^11^C]erlotinib (Appendix A), and 18.9 ± 0.8 %ID for [^11^C]elacridar (Appendix A). A decrease in the intestinal radioactivity was observed in *Abcb1a/b^(−/−)^Abcg2^(−/−)^* mice, while intestinal radioactivity in *Abcb1a/b^(−/−)^* mice and *Abcg2^(−/−)^* mice appeared to be similar to wild-type mice for all three radiotracers (Figure 4).

The PK model (Figure 1) was used to estimate the hepatic uptake clearance (CL_1_) and the rate constants defining the transfer of radioactivity from liver into the sink compartment (blood, *k*_2_) and from liver into excreted bile (*k*_3_). Visually, the model provided good fits of the observed liver and intestinal TACs (Appendix A) and parameter precision (determined as percent coefficient of variation, %CV) was acceptable (Table 2). CL_1_, *k*_2_, and *k*_3_ values were considerably lower for [^11^C]tariquidar and [^11^C]elacridar than for [^11^C]erlotinib (Table 2). No significant differences were observed for CL_1_ and *k*_2_ between wild-type and knockout mice for any of the studied radiotracers (Table 2). *k*_3_ was significantly decreased in *Abcb1a/b^(−/−)^Abcg2^(−/−)^* mice as compared to the wild-type group, by 1.96-fold for [^11^C]tariquidar, by 2.79-fold for [^11^C]erlotinib, and by 2.75-fold for [^11^C]elacridar (Figure 5). In addition, for [^11^C]erlotinib, *k*_3_ was significantly (2.11-fold) reduced in *Abcg2^(−/−)^* mice relative to wild-type mice. No significant differences were observed in *k*_3_ for [^11^C]tariquidar and [^11^C]elacridar between wild-type and single transporter knockout mice (Table 2, Figure 5).

The intestinal clearance (CL_intestine,blood_) was significantly decreased in *Abcg2^(−/−)^* and *Abcb1a/b^(−/−)^Abcg2^(−/−)^* mice as compared to the wild-type group, while it was unchanged in *Abcb1a/b^(−/−)^* mice for both [^11^C]tariquidar and [^11^C]erlotinib (Appendix A). No significant differences were observed in CL_intestine,blood_ for [^11^C]elacridar between wild-type and knockout animals (Appendix A).

## 4. Discussion

P-gp and BCRP play a protective role at the BBB by limiting the entry of xenobiotics, such as drugs, from blood into the brain. Due to their co-localization and considerable overlap in substrate specificity, P-gp and BCRP display functional redundancy in the active efflux of their common substrates across the BBB [4,6]. This was shown to lead to substantial increases in the brain distribution of dual P-gp/BCRP substrates in *Abcb1a/b^(−/−)^Abcg2^(−/−)^* mice, while only small changes were observed when only one transporter is absent (*Abcb1a/b^(−/−)^* mice and *Abcg2^(−/−)^* mice) [2,3,4,5,6]. In the present study, we used whole-body PET imaging with the i.v. administered dual P-gp/BCRP substrates [^11^C]tariquidar, [^11^C]erlotinib, and [^11^C]elacridar [11,16] in wild-type, *Abcb1a/b^(−/−)^*, *Abcg2^(−/−)^*, and *Abcb1a/b^(−/−)^Abcg2^(−/−)^* mice to assess in vivo whether a similar P-gp/BCRP functional redundancy as described for the BBB also occurs at the level of the canalicular membrane of hepatocytes.

The [^11^C]tariquidar and [^11^C]elacridar data used for this study have been previously analyzed to investigate the transport of both radiotracers by P-gp and BCRP at the mouse BBB [11]. In the previous study, the brain distribution of [^11^C]tariquidar and [^11^C]elacridar was only assessed in terms of the PET-measured brain TACs and brain-to-blood radioactivity concentration ratios at 25 min after radiotracer injection, using blood samples taken from separate groups of animals [11]. No kinetic analysis was performed to assess the brain uptake of [^11^C]tariquidar and [^11^C]elacridar. The [^11^C]erlotinib data were previously analyzed using integration plot analysis to obtain CL_uptake,brain_ as a parameter describing brain distribution of [^11^C]erlotinib in mice [16]. In order to use the same methodology, we re-analyzed the [^11^C]tariquidar and [^11^C]elacridar data using integration plot and Logan analysis to estimate CL_uptake,brain_ and *V*_T,brain_ values. To this end, an image-derived blood curve was obtained from the volume of interest placed in the left ventricle of the heart, as was previously done for [^11^C]erlotinib [16].

In agreement with the published data [11,16], only small or no increases were observed in the brain distribution (CL_uptake,brain_, *V*_T,brain_) of the three radiotracers in the single transporter knockout animals (*Abcb1a/b^(−/−)^* and *Abcg2^(−/−)^*) (Table 1, Figure 3). For both [^11^C]tariquidar and [^11^C]elacridar, brain distribution was moderately increased in *Abcb1a/b^(−/−)^* mice but not increased in *Abcg2^(−/−)^* mice, relative to wild-type mice. In contrast, for [^11^C]erlotinib, the brain distribution was to a similar extent increased in both *Abcb1a/b^(−/−)^* mice and *Abcg2^(−/−)^* mice. This suggests that P-gp plays a greater role in limiting the brain distribution of [^11^C]tariquidar and [^11^C]elacridar than that of [^11^C]erlotinib. Moreover, a greater effect of *Abcb1a/b* knockout than of *Abcg2* knockout on the brain uptake of dual P-gp/BCRP substrates in mice may be attributed to P-gp being the quantitatively more important transporter at the mouse BBB (abundance ratio of P-gp to BCRP in brain capillaries of wild-type mice: 3.9) [29]. In agreement with the expected behavior of dual P-gp/BCRP substrates [6], there was a substantial increase in CL_uptake,brain_ and *V*_T,brain_ in combination knockout mice (*Abcb1a/b^(−/−)^Abcg2^(−/−)^*) for all three radiotracers (Table 1, Figure 3). It should be noted that it cannot be excluded that apart from P-gp and BCRP additional uptake and efflux transporters influenced the brain distribution of the investigated radiotracers.

This large increase in the brain distribution of dual P-gp/BCRP substrates in *Abcb1a/b^(−/−)^Abcg2^(−/−)^* mice relative to the small changes observed in the individual transporter knockout mice is in agreement with kinetic theory [30], implying that the brain distribution of dual substrates increases as an asymptotic function of the fraction excreted (f_e_) by the transporter [4]. Thus, the sum of the increases in the brain distribution in the individual transporter knockouts will underpredict the increase observed in *Abcb1a/b^(−/−)^Abcg2^(−/−)^* mice because the total increase in brain distribution is a nonlinear function of f_e_ and not an additive parameter. The mathematical relationship describing the fold change in brain distribution (1/(1 − f_e_)) has been shown to predict the large changes in brain distribution of dual P-gp/BCRP substrates such as lapatinib in mice when both transporters were absent [4].

In order to study hepatobiliary excretion of the three radiolabeled P-gp/BCRP substrates, we analyzed the PET data with a liver PK model that was slightly modified from a previously developed model [28]. Although the [^11^C]tariquidar PET data have already been analyzed with another liver PK model [27], a re-analysis of the data was done with the modified PK model to enable a direct comparison with the other two radiotracers. In addition, integration plot analysis has previously been applied to assess the influence of P-gp and BCRP on the biliary excretion of [^11^C]erlotinib in mice [16]. Although integration plot analysis provided similar results for [^11^C]erlotinib as in this study, it does not provide a complete picture of the hepatobiliary disposition of a radiotracer [10]. Moreover, the liver PK model used in this study considers the dual blood input to the liver via the portal vein and the hepatic artery. The radiotracer concentration in the hepatic artery was assumed to be the same as the radiotracer concentration derived from the volume of interest placed in the left ventricle of the heart, while the concentration in the portal vein, which is expected to be different from the arterial concentration, was mathematically estimated as previously described [31]. In addition, the PK model assumes that radiotracer is not metabolized during the PET scan. This is supported by previous studies in wild-type mice, in which the majority of radioactivity in plasma, brain, and liver after i.v. injection of [^11^C]tariquidar, [^11^C]erlotinib, or [^11^C]elacridar was found to be in the form of unmetabolized radiotracer (percentage of unchanged radiotracer at 25–30 min after radiotracer injection, plasma: 78.3% ([^11^C]tariquidar), 90.3% ([^11^C]erlotinib), 95.8% ([^11^C]elacridar); brain: 88.8% ([^11^C]erlotinib), 95.4% ([^11^C]elacridar); liver: 92.8% ([^11^C]tariquidar), 74.8% ([^11^C]erlotinib)) [14,27,32].

In agreement with previous kinetic analysis [27], the value of *k*_3_ (representing biliary excretion) was significantly decreased for [^11^C]tariquidar in *Abcb1a/b^(−/−)^Abcg2^(−/−)^* mice, while it remained unchanged in both *Abcb1a/b^(−/−)^* mice and *Abcg2^(−/−)^* mice (Figure 5, Table 2). Similarly, [^11^C]elacridar showed minimal changes in *k*_3_ in both individual transporter knockout mouse groups as compared to wild-type mice, while *k*_3_ was significantly decreased in *Abcb1a/b^(−/−)^Abcg2^(−/−)^* mice. This suggests that, similar to [^11^C]tariquidar, the biliary excretion of [^11^C]elacridar is mediated by both P-gp and BCRP. Accordingly, *k*_3_ for [^11^C]erlotinib was also significantly decreased in *Abcb1a/b^(−/−)^Abcg2^(−/−)^* mice, similar to the previously reported reduction in *k*_bile_ estimated with integration plot analysis [16]. However, *k*_3_ was also significantly and to a similar extent reduced in *Abcg2^(−/−)^* mice as compared to wild-type mice, while no significant changes were observed for *k*_3_ in *Abcb1a/b^(−/−)^* mice (Figure 5, Table 2). This suggests that although both P-gp and BCRP contribute to the efflux of [^11^C]erlotinib at the mouse BBB, BCRP mainly mediates the biliary excretion of [^11^C]erlotinib in the mouse liver, with only a minor contribution of P-gp. These differences among the radiotracers may be ascribed to different affinities (K_m_) to and transport capacities (V_max_) by P-gp and BCRP. A detailed assessment would require the performance of comparative in vitro saturation experiments in transporter-overexpressing cell lines.

The same kinetic theory used for the brain distribution of dual P-gp/BCRP substrates can be applied to predict the fold change in biliary excretion of dual substrates [30]. Thus, the loss of function of only one transporter will have minor consequences, while the loss of function of both transporters will lead to an exponential decrease in biliary excretion. This may have direct consequences for the risk of occurrence of transporter-mediated drug–drug interactions (DDIs) for dual P-gp/BCRP substrate drugs at the level of the canalicular hepatocyte membrane [1]. Similar to the brain [4], significant changes in hepatic drug disposition are only expected to occur when both transporters are simultaneously inhibited. As few clinically used drugs are likely to inhibit both P-gp and BCRP simultaneously, it appears rather unlikely that dual P-gp/BCRP substrate drugs become victims of P-gp- and BCRP-mediated DDIs in the liver, due to the functional redundancy between these two transporters. For all three investigated radiotracers, the fold decrease in biliary excretion was not as great as the fold increase in their brain uptake in *Abcb1a/b^(−/−)^Abcg2^(−/−)^* mice, indicating that the combined fractions of radiotracer excreted by P-gp and BCRP at the canalicular membrane of hepatocytes are not much greater than 0.5 [30]. This may be potentially related to an additional involvement of other transporters in the biliary excretion of the investigated radiotracers.

An alternative explanation for the only moderate changes in brain uptake or biliary excretion of the dual P-gp/BCRP substrates in single transporter knockout mice could be a compensatory upregulation of P-gp in *Abcg2^(−/−)^* mice or an upregulation of BCRP in *Abcb1a/b^(−/−)^* mice. However, it has been shown before by means of quantitative proteomics that there are no changes in the abundance of P-gp, BCRP, and other ABC and solute carrier (SLC) transporters in brain capillary endothelial cells of the same knockout mouse models as used in this study [33]. This supports that the functional redundancy between P-gp and BCRP observed at the mouse BBB is not related to compensatory changes in transporter expression. On the other hand, quantitative proteomics data for transporter expression in the liver of the investigated transporter knockout mouse models are currently not available. One study showed that no overexpression of *Abcg2* mRNA occurs in the liver of the same *Abcb1a/b^(−/−)^* mouse model as used in this study [34]. However, mRNA levels do not always correlate with protein abundance, and highly expressed proteins may not require high expression levels of mRNA. In addition, mRNA expression of *Abcb1a* and protein expression of P-gp (ABCB1A) in the liver of the *Abcg2^(−/−)^* mouse model have been investigated [35]. While *Abcb1a* mRNA expression was unchanged in *Abcg2^(−/−)^* mice, there was a significant, approximately twofold increase in the hepatic abundance of ABCB1A protein in *Abcg2^(−/−)^* mice. This could imply that compensatory P-gp upregulation may have at least partly contributed to the lack of changes in *k*_3_ of [^11^C]tariquidar and [^11^C]elacridar in *Abcg2^(−/−)^* mice. Thus, although similar effects as observed at the BBB were observed in the biliary excretion of dual P-gp/BCRP substrates, suggesting functional redundancy between these two transporters at the canalicular membrane of hepatocytes, a contribution of compensatory transporter upregulation cannot be entirely discarded.

It further remains to be determined whether similar effects as in mice occur in human hepatocytes, which differ from rodents in terms of the relative abundances of different membrane transporters [36]. PET studies with radiolabeled P-gp/BCRP substrates may be helpful to answer these questions in humans [27]. The upcoming availability of total-body PET scanners with large axial fields of view of up to 200 cm [37] will provide the unprecedented possibility to simultaneously assess the effects of transporters on drug disposition in humans in several different organs, such as the brain and the liver, or in primary tumors and metastases.

## 5. Conclusions

In this study, we evaluated whether the known functional redundancy between P-gp and BCRP at the BBB also extends to the canalicular membrane of hepatocytes, where both transporters mediate the biliary excretion of drugs and their metabolites. To this end, we performed PET scans with the three metabolically stable dual P-gp/BCRP substrates [^11^C]tariquidar, [^11^C]erlotinib, and [^11^C]elacridar in wild-type, *Abcb1a/b^(−/−)^*, *Abcg2^(−/−)^*, and *Abcb1a/b^(−/−)^Abcg2^(−/−)^* mice. Using different PK analysis approaches, we observed that brain distribution and biliary excretion of the three studied radiotracers was markedly changed in *Abcb1a/b^(−/−)^Abcg2^(−/−)^* mice as compared to wild-type mice, while only moderate changes were observed when only one of the two transporters was absent. Altogether, our data suggest functional redundancy between P-gp and BCRP both in controlling the brain distribution and the biliary excretion of dual P-gp/BCRP substrates, which may point to a low risk for the occurrence of transporter-mediated DDIs. Our study highlights the potential of PET imaging for assessing the effects of transporters on drug disposition on a whole-body level.

## Figures and Tables

**Figure 1 pharmaceutics-13-01286-f001:**
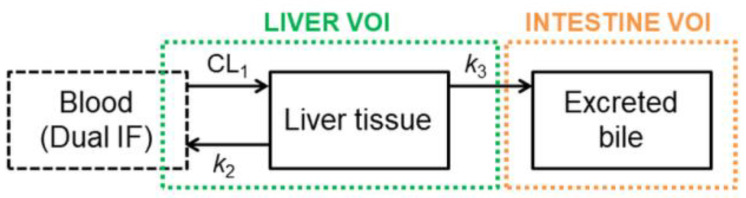
PK model used to assess the hepatobiliary disposition of [^11^C]tariquidar, [^11^C]erlotinib, and [^11^C]elacridar. CL_1_ (mL/min) represents the hepatic uptake clearance, and *k*_2_ and *k*_3_ (min^−1^) are the rate constants describing the transfer of radioactivity from liver to the sink compartment (blood) and from liver to excreted bile (intestine volume of interest), respectively. The model was modified from a previously published model (Adapted from [28], AAPS, 2019). IF: input function; VOI: volume of interest.

**Figure 2 pharmaceutics-13-01286-f002:**
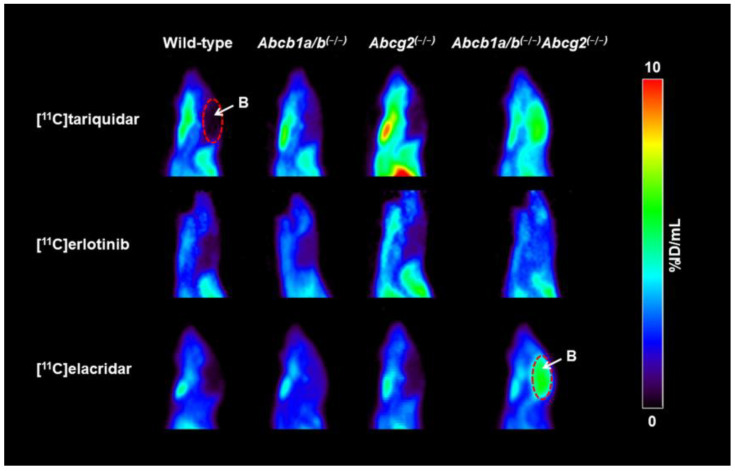
Sagittal PET summation images (0–60 min) of the head obtained after i.v. injection of either [^11^C]tariquidar, [^11^C]erlotinib, or [^11^C]elacridar in one representative wild-type, *Abcb1a/b^(−/−)^*, *Abcg2^(−/−)^*, and *Abcb1a/b^(−/−)^Abcg2^(−/−)^* mouse. Radioactivity concentration is expressed as percent of injected dose per milliliter (%ID/mL). Brain (B) is labeled with white arrows and red dashed-line areas.

**Figure 3 pharmaceutics-13-01286-f003:**
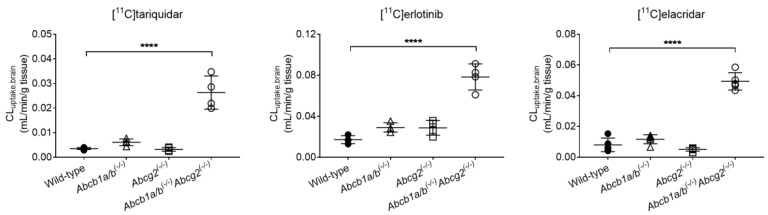
Initial brain uptake clearance (CL_uptake,brain_) of [^11^C]tariquidar, [^11^C]erlotinib, and [^11^C]elacridar determined with integration plot analysis in wild-type, *Abcb1a/b^(−/−)^*, *Abcg2^(−/−)^*, and *Abcb1a/b^(−/−)^Abcg2^(−/−)^* mice. **** *p* ≤ 0.0001, one-way ANOVA followed by a Dunnett’s multiple comparison test against the reference group (wild-type).

**Figure 4 pharmaceutics-13-01286-f004:**
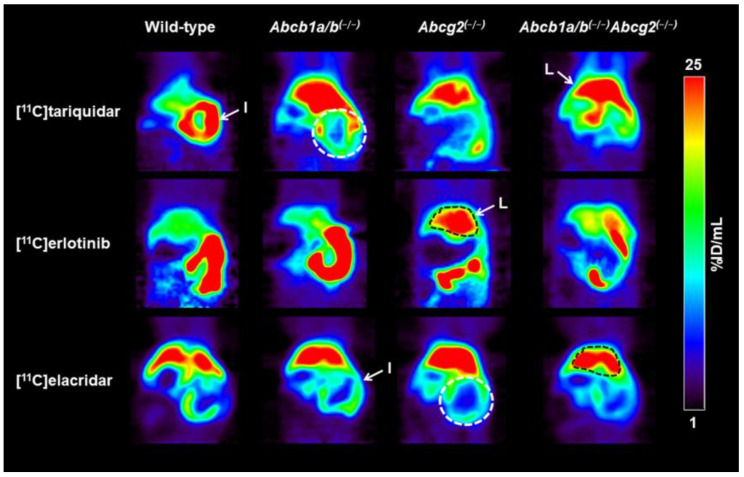
Coronal PET summation images (0–60 min) of the abdominal region obtained after i.v. injection of either [^11^C]tariquidar, [^11^C]erlotinib, or [^11^C]elacridar in one representative wild-type, *Abcb1a/b^(−/−)^*, *Abcg2^(−/−)^*, and *Abcb1a/b^(−/−)^Abcg2^(−/−)^* mouse. Radioactivity concentration is expressed as percent of injected dose per milliliter (%ID/mL). Anatomical structures are labeled with white arrows and colored dashed-line areas (I: intestine, white; L: liver, black).

**Figure 5 pharmaceutics-13-01286-f005:**
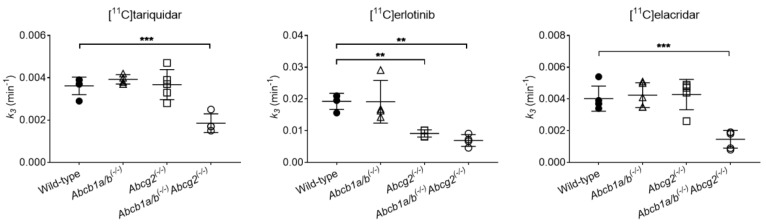
Rate constant (*k*_3_) representing the transfer of radioactivity from liver to excreted bile (intestinal region of interest) obtained with the compartmental model for [^11^C]tariquidar, [^11^C]erlotinib, and [^11^C]elacridar in wild-type, *Abcb1a/b^(−/−)^*, *Abcg2^(−/−)^*, and *Abcb1a/b^(−/−)^Abcg2^(−/−)^* mice. ** *p* ≤ 0.01, *** *p* ≤ 0.001, one-way ANOVA followed by a Dunnett’s multiple comparison test against the reference group (wild-type).

**Table 1 pharmaceutics-13-01286-t001:** PK parameters obtained with graphical non-compartmental analysis describing the brain distribution of [^11^C]tariquidar, [^11^C]erlotinib, and [^11^C]elacridar in wild-type, *Abcb1a/b^(−/−)^*, *Abcg2^(−/−)^*, and *Abcb1a/b^(−/−)^Abcg2^(−/−)^* mice.

Radiotracer	Parameter	Wild-Type	*Abcb1a/b^(−/−)^*	*Abcg2^(−/−)^*	*Abcb1a/b^(−/−)^Abcg2^(−/−)^*
[^11^C]tariquidar	CL_uptake,brain_ (mL/min/g tissue)	0.0035 ± 0.0004	0.0061 ± 0.0014	0.0031 ± 0.0007	0.0263 ± 0.0068 *
	*V* _T,brain_	0.1269 ± 0.0133	0.2169 ± 0.0227	0.1045 ± 0.0065	0.7460 ± 0.1504 *
[^11^C]erlotinib	CL_uptake,brain_ (mL/min/g tissue)	0.0171 ± 0.0038	0.0290 ± 0.0045	0.0287 ± 0.0071	0.0784 ± 0.0128 *
	*V* _T,brain_	0.2016 ± 0.0141	0.2622 ± 0.0079	0.2335 ± 0.0174	0.5015 ± 0.0497 *
[^11^C]elacridar	CL_uptake,brain_ (mL/min/g tissue)	0.0080 ± 0.0044	0.0116 ± 0.0030	0.0051 ± 0.0011	0.0493 ± 0.0564 *
	*V* _T,brain_	0.1547 ± 0.0489	0.2460 ± 0.0673	0.1097 ± 0.0315	0.6048 ± 0.2069 *

Data are given as mean ± SD (*n* = 4–5 per group). CL_uptake,brain_: initial brain uptake clearance estimated with integration plot analysis; *V*_T,brain_: total brain distribution volume estimated with Logan graphical analysis. * *p* ≤ 0.0001, one-way ANOVA followed by a Dunnett’s multiple comparison test against the wild-type group.

**Table 2 pharmaceutics-13-01286-t002:** PK parameters obtained with the compartmental model describing the hepatobiliary disposition of [^11^C]tariquidar, [^11^C]erlotinib, and [^11^C]elacridar in wild-type, *Abcb1a/b^(−/−)^*, *Abcg2^(−/−)^*, and *Abcb1a/b^(−/−)^Abcg2^(−/−)^* mice.

Radiotracer	Parameter	Wild-Type	*Abcb1a/b^(−/−)^*	*Abcg2^(−/−)^*	*Abcb1a/b^(−/−)^Abcg2^(−/−)^*
[^11^C]tariquidar	CL_1_ (mL/min)	2.2798 ± 0.5058 (12.9–32.3)	1.7207 ± 0.1377 (10.7–23.1)	1.5836 ± 0.1993 (9.6–22.1)	1.9316 ± 0.6933 (9.6–34.2)
	*k*_2_ (min^−1^)	0.2600 ± 0.0475 (11.0–26.4)	0.2421 ± 0.0227 (10.5–19.0)	0.3244 ± 0.0204 (9.0–20.5)	0.3441 ± 0.1549 (9.0–29.7)
	*k*_3_ (min^−1^)	0.0036 ± 0.0004 (4.1–7.6)	0.0039 ± 0.0002 (4.2–7.6)	0.0037 ± 0.0007 (2.8–9.1)	0.0019 ± 0.0004 * (5.3–8.4)
[^11^C]erlotinib	CL_1_ (mL/min)	5.9437 ± 2.4249 (5.3–51.5)	5.2978 ± 1.8375 (12.1–45.4)	6.3856 ± 1.7690 (10.6–33.4)	7.3708 ± 0.2309 (20.7–36.7)
	*k*_2_ (min^−1^)	0.6431 ± 0.2567 (5.1–47.3)	0.5815 ± 0.1818 (10.8–41.4)	0.6051 ± 0.0664 (10.7–30.8)	0.5484 ± 0.1021 (17.6–32.2)
	*k*_3_ (min^−1^)	0.0193 ± 0.0025 (1.2–4.2)	0.0191 ± 0.0067 (1.7–4.7)	0.0091 ± 0.0011 * (1.9–3.3)	0.0069 ± 0.0018 * (2.4–4.1)
[^11^C]elacridar	CL_1_ (mL/min)	0.5338 ± 0.1759 (6.0–14.3)	0.6887 ± 0.1939 (11.3–22.4)	0.4304 ± 0.1392 (4.0–21.0)	0.5777 ± 0.2241 (4.9–24.6)
	*k*_2_ (min^−1^)	0.1121 ± 0.0641 (7.7–13.5)	0.1872 ± 0.1165 (10.5–17.5)	0.1674 ± 0.1366 (4.8–20.6)	0.1426 ± 0.0728 (7.1–18.8)
	*k*_3_ (min^−1^)	0.0040 ± 0.0008 (4.0–6.8)	0.0042 ± 0.0008 (5.2–8.9)	0.0043 ± 0.0010 (3.2–18.6)	0.0015 ± 0.0006 * (7.0–23.3)

Data are given as mean ± SD (*n* = 4–5 per group). Values in parentheses represent the range in percent coefficient of variation (%CV), which determines the parameter precision. PK parameters were obtained with the liver PK model. CL_1_ defines the hepatic uptake clearance, and *k*_2_ and *k*_3_ are the rate constants defining the transfer of radioactivity from liver to the sink compartment (blood) and from liver to excreted bile, respectively. * *p* ≤ 0.01, one-way ANOVA followed by a Dunnett’s multiple comparison test against the wild-type group.

## Data Availability

The data presented in this study are available on request from the corresponding author.

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
