# Peer review of "Assessing the Functional Redundancy between P-gp and BCRP in Controlling the Brain Distribution and Biliary Excretion of Dual Substrates with PET Imaging in Mice"

_pharmaceutics, 2021, doi:10.3390/pharmaceutics13081286_

Round 1
Reviewer 1 Report
This manuscript reports the results on assessing the functional redundancy between P-gp and BCRP in controlling the brain distribution and biliary excretion of dual substrates with PET imaging in mice. The experimental design seems to be reasonable and obtained data looks interesting. However, there are some suggestions and/or question to be improved. Please consider the below comments.
- Title: the term of “redundancy” is not likely defined to provide the authors’ conclusion in the manuscript. Is there any previous report to define “redundancy” term? If not, “interplay” term may be better.
- Introduction section: For the test drugs, [11C]tariquidar, [ 11C]erlotinib and [11C]elacridar as dual substrates for P-gp and BCRP, the biliary clearance information is needed in detail. How much the biliary recovery of these drugs are reported in mouse? Also, negligible metabolism information of these drugs are described more detail with experimental values in terms of metabolic clearance.
- The major concern is about the assumption of no direct secretion from blood into the intestine occurred after intravenous injection in this study. There was no scientific information whether this assumption is quite correct or not. Please provide the scientific information or previous data to present the intestinal secretion of these drugs are negligible. Normally, there is intestinal secretion for P-gp or BCRP substrates after IV injection.
- In the Table 2, only erlotinib showed significant changes of k3 parameter for Abcg knockout mouse and Abcb1 and Abcg double knockout mouse, not Abcb1 knockout mouse. This phenomena did not exist for tariquidar, and elacridar. Please discuss this important result based on the parameters such as Km, Vmax and intrinsic clearance (CLint) of P-gp or BCRP transporter for tariquidar, erlotinib and elacridar. The different affinity and/or capacity for P-gp or BCRP should be considered.
- About the equation 1, K uptake, brain has unit of clearance, so that CL uptake, brain is more accurate expression rather than K uptake, brain. Also the exact unit may be correct for mL/min/g brain, not mL/min/mL tissue. In addition, V E,brain term may refer to the capillary space and rapid adsorption/binding to the vascular surface in the brain, not initial distribution volume of brain. Please check the unit and indicated parameter based on the previous literature. (Kusuhara et al., J Pharmacol Exp Ther. 1997 Nov;283(2):574-80.).
- Is it possible to calculate apparent biliary clearance of drugs, not the elimination constant, K3?
- What is exact dose of each drug, not the radioactivity?
- For the section 3.2 for biliary excretion studies, no methodology description exists in the 2.3 section. Please add more methodological descriptions about biliary excretion studies.
Reviewer 2 Report
The manuscript entitled "Assessing the functional redundancy between P-gp and BCRP in controlling the brain distribution and biliary excretion of dual substrates with PET imaging in mice" evaluated the functional redundancy of P-gp and BCRP since both transporters locate on the BBB in human. The authors found that double KO mice showed higher brain uptake of radiotracers as well as liver to excreted bile. The radiotracers in single KO and WT mice did not change a lot. Overall, the study is interesting to the audience, while there are still some major and minor issues that need to be addressed.
1. The authors should provide information regarding the affinity or efficacy of each substrate to individual transporters. Usually, dual substrates were not equally transported by ABC transporters.
2. The authors should evaluate the effects of other ABC transporters, for example tariquidar and erlotinib were reported as inhibitors of ABCC10, another ABC transporter which is broadly expressed in human tissues.
3. The authors should provide data point regarding the abundance of both transporters in brain and liver-bile duct.
4. PET is not a new technique, also it's been used in locating radiotracers in previous studies. Thus evaluating the feasibility of PET is not innovative.
Reviewer 3 Report
General Comments:
The study by Hernández-Lozano et al. tested positron emission tomography (PET) using the 11C-labeled P-gp/BCRP substrates [11C]tariquidar, [11C]erlotinib and [11C]elacridar to evaluate whether the known functional redundancy between P-gp and BCRP at the blood-brain barrier (BBB) also exists in the liver, where both transporters mediate the biliary excretion of drugs. To this end, the authors performed whole-body [11C]tariquidar, [11C]erlotinib and [11C]elacridar PET in Wild-type, Abcb1a/b(-/-), Abcg2(-/-) and Abcb1a/b(-/-)Abcg2(-/-) mice and investigated both the brain distribution and the biliary excretion of the three tracers qualitatively and quantitatively in the four mouse strains.
The authors described that brain distribution and biliary excretion of the three radiotracers was markedly increased in Abcb1a/b(-/-)Abcg2(-/-) mice as compared to wild-type mice. While only moderate changes were observed when only one of the two transporters was absent. They concluded that functional redundancy exists between P-gp and BCRP in controlling both the brain distribution and biliary excretion of dual P-gp/BCRP substrates and highlighted the utility of PET as an upcoming tool to assess the effect of transporters on drug disposition on a whole-body level.
Several studies have provided evidence for functional redundancy between P-glycoprotein (P-gp) and breast cancer resistance protein (BCRP) in limiting the brain distribution of dual P-gp/BCRP substrates. In addition, both membrane transporters have a broad and largely overlapping substrate spectrum including a variety of clinically used drugs, such as most molecularly targeted anticancer drugs. Therefore, the topic addressed by the present manuscript is of interest not only in the field of drug development for cancer therapy but also for molecular imaging using PET.
Although, the study is relatively well designed, there are some critical issues in this report that need to be addressed carefully prior to be recommended for publication in Pharmaceutics. In particular, the data presentation and the key results on the brain uptake and the hepatobiliary excretion of the radiotracers (from PET imaging) as well as part of the Discussion are not convincing or are not accurately described.
In details:
- The potential clinical impact of the functional redundancy between P-gp and BCRP in tumor therapy, especially in breast cancer as well as the rolle of PET imaging using radiolabeled P-gp and BCRP are poorly described in the introduction and are not accurately discussed. This is particularly important and would increase the scientific and clinical impact of this work significantly.
- Most of the data presented in this report, including the radiosynthesis, data sets and PET imaging have been already published by the same group. It is really difficult to identify the novelty in this work.
- The authors stated that the transfer of radioactivity from liver to excreted bile was significantly lower in Abcb1a/b(-/-)Abcg2(-/-) mice and almost unchanged in Abcb1a/b(-/-) and Abcg2(-/-) mice, with the exception of [11C]erlotinib for which biliary excretion was also significantly reduced in Abcg2(-/-) mice. How could the authors explain this discrepancy. This issue needs to be addressed and discussed accurately.
- The Discussion in the present paper is not always informative and in part relatively hard to understand. The key results obtained from the present work, which led to the conclusion that a functional redundancy may exists between P-gp and BCRP in controlling the brain distribution and biliary excretion of P-gp/BCRP substrates should be discussed more accurately.
- The discussion related to the features of PET as a prediction tool to assess the effect of transporters on drug disposition on the whole-body, especially in oncological patients with breast cancer is lacking. This is particularly important and would increase the scientific and clinical impact of this work significantly.
Specific remarks:
- The English language in the manuscript is relatively variable. Please cross-check the manuscript for spelling and grammatical errors.
- The presentation of the results in some figures and tables could be improved. In particular, Figure 2 and Fig. 4 are confusing and difficult to understand. The differences in the uptake in the targeted organs in Abcb1a/b(-/-)Abcg2(-/-) mice as compared to wild-type mice are not remarkable or at least not convincing on the images. Another quantification method e.g. in form of table or column should probably be preferable.
- Please check the cited literatures in the text.
Round 2
Reviewer 1 Report
The revised manuscript tries to reflect the reviewer’s concerns and/or suggestion to some extent. However, there are a minor point to be cleared for the pharmacokinetic parameter, k uptake, brain. Although the authors made some response, k uptake, brain parameter has a unit for mL/min/mL tissue indeed.
By integration plot analysis, the brain uptake clearance (CLuptake, brain) is calculated by the initial slope of a plot of Xt,brain/Ct, blood versus AUC0-t/Ct,blood. Namely, CLuptake,brain has a unit mL/min/g brain in the many previous literatures as below.
P-Glycoprotein mediates the efflux of quinidine across the blood-brain barrier. H Kusuhara 1, H Suzuki, T Terasaki, A Kakee, M Lemaire, Y Sugiyama J Pharmacol Exp Ther. 1997 Nov;283(2):574-80.
Role of P-glycoprotein in tissue uptake of indinavir in rat. Mehrdad Hamidi. Life Sci. 2006 Aug 1;79(10):991-8.
Developmental changes in P-glycoprotein function in the blood-brain barrier of nonhuman primates: PET study with R-11C-verapamil and 11C-oseltamivir. Tadayuki Takashima 1, Chihiro Yokoyama, Hiroshi Mizuma, Hajime Yamanaka, Yasuhiro Wada, Kayo Onoe, Hiroko Nagata, Shusaku Tazawa, Hisashi Doi, Kazuhiro Takahashi, Masataka Morita, Motomu Kanai, Masakatsu Shibasaki, Hiroyuki Kusuhara, Yuichi Sugiyama, Hirotaka Onoe, Yasuyoshi Watanabe J Nucl Med. 2011 Jun;52(6):950-7.
Involvement of organic anion transporters in the efflux of uremic toxins across the blood-brain barrier. Tsuneo Deguchi 1, Kouya Isozaki, Kouno Yousuke, Tetsuya Terasaki, Masaki Otagiri. J Neurochem. 2006 Feb;96(4):1051-9
In addition, the first rate constant cannot have a unit to indicate blood(plasma) volume cleared per min. The first rate constant unit should be like per min.
Thus, the reviewer sincerely would like to suggest to correct k uptake, brain by CLuptake, brain in the revised manuscript to follow the previous representative literatures.
Author Response
As suggested by the reviewer and taking into account the mentioned literature, we have replaced the term kuptake,brain by CLuptake,brain throughout the text as well as in Figure 3 and Table 1.
Reviewer 2 Report
The authors have addressed my concerns.
Author Response
Thank you.
Reviewer 3 Report
I have reviewed the revised version of the manuscript "Assessing the functional redundancy between P-gp and BCRP 2 in controlling the brain distribution and biliary excretion of 3 dual substrates with PET imaging in mice" (MS ID: pharmaceutics-1322934) by Irene Hernández-Lozano et al. for potential publication in Pharmaceutics.
The authors have addressed all issues and suggestions accurately and point by point. In particular, the introduction has been rewritten and important results of the study discussed more accurately in detail. The manuscript has been sufficiently improved to warrant publication in Pharmaceutics and can be recommended in its present form for publication in Pharmaceutics.
Author Response
Thank you.